# EUROlinkCAT protocol for a European population-based data linkage study investigating the survival, morbidity and education of children with congenital anomalies

Joan K Morris ,[1] Ester Garne ,[2] Maria Loane ,[3] Ingeborg Barisic ,[4] James Densem,[5] Anna Latos-Bieleńska ,[6] Amanda Neville,[7] Anna Pierini ,[8] Judith Rankin ,[9] Anke Rissmann ,[10] Hermien de Walle ,[11] Joachim Tan ,[1] Joanne Emma Given ,[3] Hugh Claridge ,[1] EUROlinkCAT Consortium

► Prepublication history and supplemental material for this paper is available online. To view these files, please visit the journal online (http://dx.doi.org/10.1136/bmjopen-2020-047859).

For numbered affiliations see end of article.

**Correspondence to**
Professor Joan K Morris; jmorris@sgul.ac.uk

## ABSTRACT

**Introduction** Congenital anomalies (CAs) are a major cause of infant mortality, childhood morbidity and long-term disability. Over 130 000 children born in Europe every year will have a CA. This paper describes the EUROlinkCAT study, which is investigating the health and educational outcomes of children with CAs for the first 10 years of their lives.

**Methods and analysis** EUROCAT is a European network of population-based registries for the epidemiological surveillance of CAs. EUROlinkCAT is using the EUROCAT infrastructure to support 22 EUROCAT registries in 14 countries to link their data on births with CAs to mortality, hospital discharge, prescription and educational databases. Once linked, each registry transforms their case data into a common data model (CDM) format and they are then supplied with common STATA syntax scripts to analyse their data. The resulting aggregate tables and analysis results are submitted to a central results repository (CRR) and meta-analyses are performed to summarise the results across all registries. The CRR currently contains data on 155 594 children with a CA followed up to age 10 from a population of 6 million births from 1995 to 2014.

**Ethics** The CA registries have the required ethics permissions for routine surveillance and transmission of anonymised data to the EUROCAT central database. Each registry is responsible for applying for and obtaining additional ethics and other permissions required for their participation in EUROlinkCAT.

**Dissemination** The CDM and associated documentation, including linkage and standardisation procedures, will be available post-EUROlinkCAT thus facilitating future local, national and European-level analyses to improve healthcare. Recommendations to improve the accuracy of routinely collected data will be made.

Findings will provide evidence to inform parents, health professionals, public health authorities and national treatment guidelines to optimise diagnosis, prevention and treatment for these children with a view to reducing health inequalities in Europe.

### Strengths and limitations of this study

► The implementation of a common data model enables the same centrally developed syntax script to be run in all registries which is efficient and ensures standardisation of analysis.
► The use of a reference population allows country differences to be adjusted for and enables more accurate comparisons of the burden of disease attributable to congenital anomalies (CAs) across countries to be made.
► Merging births with CAs to their records in routine healthcare data enables a detailed evaluation of the accuracy of the data and enables improvements to be suggested.
► Being unable to share individual case data or aggregate data that might be disclosive reduces the power of the analysis that can be performed, particularly for rare anomalies.
► Only specific areas in Europe are represented, with a lack of data in particular from Eastern Europe.

## INTRODUCTION

Over 130 000 children born in Europe every year will have a major congenital anomaly (CA), equivalent to 2.5% of all European births. CAs include structural defects, chromosomal anomalies and genetic syndromes. CAs are a leading cause of perinatal and infant mortality, especially in developed countries.[1] From 2003 to 2012, CAs were associated with about 40% of all infant deaths in Sweden and England.[2] There is a large variation in child death rates across Europe; in 2013 the child death rates (age 0–14 years) were 60% higher in the UK and Belgium compared with Sweden, with an additional 10 countries being 30% higher than Sweden.[3]

To identify potentially preventable and remedial causes it is important to investigate the health inequalities in survival in children with CA across Europe.

It has been shown that relying on death certificates as a source of information on mortality due to CAs does not provide an accurate assessment of the survival for children with specific CAs. Death certificates state the direct or primary cause of death which may be infection, seizures or others and therefore may not mention the CA.[4] Copeland and Kirby[4] concluded that the only way to accurately study mortality and survival in children with rare CAs is to pool data across CA registries and link these to death registries.

Advances in fetal, neonatal and paediatric care have improved outcomes for individuals with some CAs, for example, Down syndrome (DS)[5–7] and cardiac anomalies. Several studies have shown that children with CAs account for a very high proportion of all hospital admissions.[8 9] However, there is a lack of information on the length of hospital stays for children with specific CAs, with most studies concerning children with DS, orofacial clefts or congenital heart defects (CHD).[10] Often hospital stays are investigated for the first 2 or 3 years of a child's life.[11–14] However, Wehby et al showed that hospital admissions for those born with orofacial clefts were increased at all ages up to 60 years of age.[15] Rarely has length of hospital stay been related to other factors, such as social class. Two studies (Derrington 2013 in the USA to 3 years of age and Hung 2011 in Taiwan for all ages) both identified other factors such as ethnicity and socioeconomic factors as important influences on the length of inpatient stays in children and adults with DS.[12 16]

The proportion of children born with a CA surviving beyond infancy is increasing.[5 6] How these children are performing in school and their additional educational needs is therefore becoming increasingly important as there may be a growing population of children and young people requiring additional support and resources in the future. However, apart from the more common genetic syndromes, there is a paucity of information about this.[17 18] The American Heart Association reviewed the literature on children with CHD and concluded that they are at an increased risk of developmental delay, even once the frequent occurrence of genetic syndromes has been taken into account, particularly for neonates or infants requiring open heart surgery.[19] Wehby et al also showed that children with isolated orofacial clefts were at a much greater risk of low achievement at school than their classmates.[20] A systematic review of neurocognitive outcomes following general anaesthesia and surgery in children concluded that exposure to general anaesthesia in young children did affect their development in some neurocognitive domains.[21] However, the authors recommended that the effects of surgery should be considered separately for each specific anomaly.

EUROCAT (https://eu-rd-platform.jrc.ec.europa.eu/eurocat) is a European network of population-based CA registries, which started in 1979 and has expanded to include 39 registries in 21 countries covering more than 29% of European births (1.7 million) per year.[22–24] The main objectives of EUROCAT are to provide essential epidemiological information and surveillance on CAs in Europe, to evaluate the effectiveness of primary prevention and to assess the impact of developments in prenatal screening.[25 26] Hence, the emphasis is on information collected up to a baby's first year of life.

The aim of EUROlinkCAT is to investigate the survival, morbidity and educational outcomes of children with specific CAs for the first 10 years of their lives by linking births with CAs in EUROCAT registries to electronic healthcare and education databases. The availability of population-based data on births with CAs across the EUROCAT network will enable survival, morbidity and education to be investigated for specific CAs as well as differences in these outcomes across Europe according to specific risk factors and social inequalities to be explored.

Electronic healthcare data are increasingly being used by researchers to investigate the epidemiology of CAs, rather than using information from CA registries. Such healthcare data have often been found to be incomplete.[27–31] A small number of registries will analyse the maternal pregnancy records for women registered as having had a termination of pregnancy for a fetal anomaly (TOPFA) in EUROCAT. This will enable the accuracy of routine information on TOPFAs to be evaluated. The accuracy of CA coding in live births will be evaluated by comparing the EUROCAT data for live births with the CA diagnosis from the electronic healthcare databases covering both inpatient and outpatient visits. The information on death certificates will be compared with the anomalies recorded in EUROCAT. Recommendations will be developed to enable the maximum information from electronic healthcare data to be extracted for research purposes and to quantify the amount of data that cannot be obtained.

This paper describes the design of the study, the methods used to obtain and analyse the linked data and evaluates the first 3 years' progress of EUROlinkCAT.

## METHODS AND ANALYSIS
### Design and setting
In 2017, all EUROCAT registries were invited to participate in the EUROlinkCAT study. Twenty-two registries from 14 countries agreed to participate and to link all live births with a CA registered in their registries and born from 1 January 1995 to 31 December 2014. Almost all EUROCAT registries send anonymised data on CAs occurring in all live births, fetal deaths from 20 weeks gestation and TOPFAs to the EUROCAT central database. Comprehensive coding instructions[32] and the use of the EUROCAT Data Management Programme (EDMP) to import data into the central database ensure that standard variables, definitions and coding are used by all registries in the network. CAs are coded locally using the WHO International Statistical Classification of Diseases

and Related Health Problems (ICD) 9th or 10th Revision with the British Paediatric Association code extension offering more specificity (table 1). Cases with minor anomalies only are excluded (see EUROCAT Guide 1.4, Minor Anomalies for Exclusion (V.14.10.14)). Registries can code up to nine anomalies for each case and provide additional information in the specified text fields. Based on the ICD-9 or ICD-10 codes present, cases are automatically assigned by EDMP to defined major CA subgroups in accordance with the EUROCAT Guide 1.4. A case with more than one major anomaly may be assigned to more than one subgroup. Since 2015, the central database has been hosted by the European Commission Joint Research Centre in Ispra (Italy).

Tables 1 and 2 provide a list of the 81 EUROlinkCAT CA subgroups which include structural anomalies, genetic syndromes and chromosomal anomalies that will be investigated. There are 60 EUROCAT subgroups (table 1) and an additional 21 new CA subgroups not defined in EUROCAT (table 2). The subgroups have been identified as being reasonably homogeneous to provide meaningful information and also to be prevalent enough to enable sufficiently precise estimates to be obtained from the analyses. For example, the EUROCAT subgroup 'Chromosomal' was not included as it includes all genetic syndromes, but specific syndromes such as DS (a EUROCAT subgroup) and Di George syndrome (a new subgroup) were included. For some analysis, such as mortality, DS children will be analysed according to whether they have a cardiac anomaly and/or a gastrointestinal anomaly, as these are common and are likely to influence their survival. As outcomes are expected to be more severe for children with multiple and more complex CAs, analyses are also performed separately for children with isolated anomalies or with multiple anomalies defined according to the methodology by Garne *et al*.[33] Isolated anomalies are defined as a CA in one organ system only or with a known sequence where multiple CAs cascade as a consequence of a single primary anomaly. Multiple anomalies are defined as two or more major structural CAs in different organ systems, where the pattern of anomalies has not been recognised as part of a syndrome or sequence.

## Linkage

Table 3 provides details of the linkages originally planned by the EUROCAT registries and the current linkages occurring (as of August 2020). The reasons why some registries could not link their data are explored in detail in another paper submitted for publication—they include not being able to obtain the necessary permissions, relevant outcomes not being recorded in specific data sources and the time scale for the data supply being after the end of the study's funding. Currently, 19 registries are linking their data to mortality records, 15 plan to link to hospital in-patient records and 7 to prescription records for the work package that will consider morbidity for children born with a CA. At the time of writing, nine registries plan

to link their information on children with CAs to education records. To evaluate the accuracy of the routine healthcare data, five registries are additionally linking to outpatient data and four will also link to pregnancy information recorded in the mother's health records about TOPFAs. The 19 registries survey over 6 million births in the population.

For the evaluation of survival and morbidity, each child will be followed up for a maximum of 10 years. This age cut-off has been chosen to enable enough children to be identified and followed up; a longer follow-up would mean fewer children would be eligible as currently national or local electronic healthcare record sources often do not go back more than 10 years. For education the maximum follow-up is until the end of compulsory school age (typically 16 across participating countries), although for some registers data are only available for a shorter period of follow-up. The longer follow-up was chosen, because in Finland there are no national education tests and national data on education attainment are available at the age of 15–16 (9th grade).

## Reference population

Where possible, each registry obtained information from electronic healthcare records, prescription records and education records on children without a CA. The definition of these 'control' cohorts will vary according to the registry, ranging from all children in the same population covered by the registry to a 10% random sample of children stratified by birth year and child's sex. The use of such a reference population is essential in interpreting differences across countries, as it will provide information on key outcomes, such as duration of hospital stays and medication prescribing, on children without reported anomalies, which is expected to vary by country. Table 4 provides information on the reference populations being identified.

## Standardisation and common data models

EUROCAT registries submit 96 core and non-core variables to the EUROCAT central database providing pseudonymised information on the baby and mother, diagnosis, karyotype (if known), exposure, family history and sociodemographic details. These have already been standardised and table 5 lists the 52 variables of which 34 are core variables and their common coding scheme that are used in the EUROlinkCAT study. In contrast, all the data obtained from linkage have to be standardised to a common format, as the healthcare and educational systems across Europe use different native languages and coding classification schemes. To do this, each registry provided their data dictionaries describing the variables in their local databases including the variable names, format, definitions and coding schemes.

Table 6 shows how the variable identifying the sex (male and female) of the child is coded in the different registries with different variable names, different formats and different coding schemes. For each substudy in

**Table 1** EUROCAT congenital anomaly subgroups in EUROlinkCAT

| EUROCAT subgroups | ICD-10-BPA | ICD-9-BPA |
|---|---|---|
| All anomalies* | Q-chapter, D215, D821, D1810†, P350, P351, P371 | 74, 75, 27910, 2281†, 76076, 76280, 7710, 7711, 77121 |
| **Structural anomalies** | | |
| Spina bifida | Q05 | 741 |
| Hydrocephalus | Q03 | 7423 |
| Severe microcephaly | Q02 | 7421 |
| Congenital cataract | Q120 | 74332 |
| Congenital heart defects (CHD) | Q20–Q26 | 745, 746, 7470–7474 |
| Severe CHD | Q200, Q201, Q203, Q204, Q212, Q213, Q220, Q224, Q225, Q226, Q230, Q232, Q233, Q234, Q251, Q252, Q262 | 74500, 74510, 7452, 7453, 7456, 7461, 7462, 74600, 7463, 7465, 7466, 7467, 7471, 74720, 74742 |
| Transposition of great vessels | Q203 | 74510 |
| Ventricular septal defect | Q210 | 7454 |
| Atrial septal defect (ASD) | Q211 | 7455 |
| Atrialventricular septal defect (AVSD) | Q212 | 7456 |
| Tetralogy of Fallot | Q213 | 7452 |
| Pulmonary valve stenosis | Q221 | 74601 |
| Aortic valve atresia/stenosis | Q230 | 7463 |
| Mitral valve anomalies | Q232, Q233 | 7465, 7466 |
| Hypoplastic left heart | Q234 | 7467 |
| Coarctation of aorta | Q251 | 7471 |
| Patent ductus arteriosus (PDA) as only CHD in term infants (GA+37 weeks) | Q250 | 7470 |
| Cystic adenomatous malformation of lung | Q3380 | No code |
| Cleft lip with or without cleft palate | Q36, Q37 | 7491, 7492 |
| Cleft palate | Q35 | 7490 |
| Oesophageal atresia with/without trachea-oesophageal fistula | Q390–Q391 | 75030–75031 |
| Duodenal atresia or stenosis | Q410 | 75110 |
| Atresia or stenosis of other parts of small intestine | Q411–Q418 | 75111–75112 |
| Ano-rectal atresia and stenosis | Q420–Q423 | 7521–75124 |
| Diaphragmatic hernia | Q790 | 75661 |
| Gastroschisis | Q793 | 75671 |
| Omphalocele | Q792 | 75670 |
| Multicystic renal dysplasia | Q6140, Q6141 | 75316 |
| Congenital hydronephrosis | Q620 | 75320 |
| Hypospadias | Q54 | 75260 |
| Limb reduction defects | Q71-Q73 | 7552–7554 |
| Craniosynostosis | Q750 | 75600 |
| **Chromosomal anomalies** | | |
| Down syndrome | Q90 | 7580 |
| Trisomy 13 | Q914–Q917 | 7581 |

Continued

**Table 1** Continued

| EUROCAT subgroups | ICD-10-BPA | ICD-9-BPA |
|---|---|---|
| Trisomy 18 | Q910-Q913 | 7582 |
| Turner syndrome | Q96 | 75860, 75861, 75862, 75869 |
| Klinefelter syndrome | Q980–Q984 | 7587 |
| **Rare structural anomalies with a EUROCAT subgroup** | | |
| Encephalocele | Q01 | 7420 |
| Arhinencephaly/ holoprosencephaly | Q041, Q042 | 74226 |
| Anophthalmos/microphthalmos | Q110, Q111, Q112 | 7430, 7431 |
| Anophthalmos | Q110, Q111 | 7430 |
| Congenital glaucoma | Q150 | 74320 |
| Anotia | Q160 | 74401 |
| Common arterial truncus | Q200 | 74500 |
| Double outlet right ventricle | Q201 | No code |
| Single ventricle | Q204 | 7453 |
| Triscuspid atresia and stenosis | Q224 | 7461 |
| Ebstein's anomaly | Q225 | 7462 |
| Pulmonary valve atresia | Q220 | 74600 |
| Hypoplastic right heart | Q226 | No code |
| Aortic atresia/interrupted aortic arch | Q252 | 74720 |
| Total anomalous pulmonary venous return | Q262 | 74742 |
| Choanal atresia | Q300 | 7480 |
| Hirschsprung's disease | Q431 | 75130–75133 |
| Atresia of bile ducts | Q442 | 75165 |
| Annular pancreas | Q451 | 75172 |
| Indeterminate sex | Q56 | 7527 |
| Situs inversus | Q893 | 7593 |
| VATER/VACTERL | Q8726 | 759895 |

*All anomalies=all cases of congenital anomaly, excluding cases with only minor anomalies as defined in Section 3.2 in EUROCAT Guide 1.4 for cases born post-2005. Cases with more than one anomaly are only counted once in the 'all Anomalies' subgroup.
†ICD10 code D1810 (ICD9 code 2281) is the code for cystic hygroma.
GA, gestational age; ICD-9-BPA, International Statistical Classification of Diseases and Related Health Problems 9th Revision with the British Paediatric Association; ICD-10-BPA, International Statistical Classification of Diseases and Related Health Problems 10th Revision with the British Paediatric Association.

EUROlinkCAT, a common data model (CDM) containing all variables required for its analyses were developed. All the EUROlinkCAT CDMs contain the variable L_CH_SEX, defined as 'sex of child' and with a coding scheme in integer format of 1=male, 2=female, 3=indeterminate, 9=not known '.'=not recorded or not available for study. Ulster University (UU) used the information in each registry's data dictionary to create the new EUROlinkCAT 'standardised' L_CH_SEX variable. UU, in collaboration with the registries, created registry-specific syntax scripts to standardise all the variables in the EUROlinkCAT CDMs. Online supplemental appendices 1–2 list all the variables included in the substudies (mortality, electronic healthcare records and prescription records).

The CDMs also specify how the data are stored. For mortality, all the relevant variables occur only once for each EUROCAT case and are stored in the same data file (or table) (figure 1). However, when analysing hospital admissions, each child may have more than one admission and for each admission may receive more than one diagnosis. Therefore, the hospital admissions data are stored in a separate data file (or table) from the diagnoses data and separately from the EUROCAT data on the child; each data file contains a reference key which serves to link all records belonging to one person for analysis (see figure 2). The standardisation syntax scripts from UU specify the separate data files (or tables) for each linkage containing all the variables in the CDMs.

**Table 2** New congenital anomaly subgroups in EUROlinkCAT

| New subgroups for EUROlinkCAT | ICD-10-BPA | ICD-9-BPA |
|---|---|---|
| **Structural anomalies** | | |
| Anomalies of corpus callosum | Q040 | 74221 |
| Anomalies of intestinal fixation | Q433 | 7514 |
| Unilateral renal agenesis | Q600 | No code |
| Accessory kidney | Q630 | 75330 |
| Bladder exstrophy | Q641 | 7535 |
| Epispadia | Q640 | 75261 |
| Posterior urethral valves | Q6420 | 75360 |
| Prune Belly | Q794 | 75672 |
| Arthrogryposis multiplex congenita | Q743 | 75580 |
| **Genetic syndromes** | | |
| Di George syndrome | D821 | 27910 |
| Goldenhar syndrome | Q8704 | 75606 |
| Cornelia de Lange syndrome | Q8712 | 759821 |
| Noonan syndrome | Q8714 | 759896 |
| Prader-Willi | Q8715 | 759872 |
| Beckwith-Wiedemann syndrome | Q8730 | 759874 |
| Williams syndrome | Q8784 | No code |
| Angelman syndrome | Q8785 | No code |
| **Chromosomal anomalies** | | |
| Wolff-Hirschhorn syndrome | Q933 | 75832 |
| Cri-du chat syndrome | Q934 | 75831 |
| Karyotype XXX | Q970 | 75885 |
| **Sequences** | | |
| Pierre-Robin sequence | Q8708 | 75603 |

ICD-9-BPA, International Statistical Classification of Diseases and Related Health Problems 9th Revision with the British Paediatric Association; ICD-10-BPA, International Statistical Classification of Diseases and Related Health Problems 10th Revision with the British Paediatric Association.

UU included validation routines in the syntax scripts to determine if the data have been correctly transformed. For example, it is checked that a date of death does not occur prior to the date of birth; while the primary purpose is to ensure the data have been standardised correctly, it can also reveal any errors in the linked data.

A CDM is not being defined for the linked education data as there is limited scope for comparison and pooling of data across countries. This is due in part to inherent differences in the educational stages and systems, the variability in data available, and fewer registries being able to participate (5 of the registries are from England, and 1 from Wales, Denmark, Italy and Finland).

## Work of the Standardisation Committee

In addition to defining the CDM and its structure, the EUROlinkCAT Standardisation Committee was responsible for taking other decisions, usually in consideration of local data characteristics, to ensure that data were meaningful and comparable across registries. The most important issues are listed below:

1. Inclusion according to gestational age (GA) at birth: for the mortality study all live births with a GA below 24 weeks were excluded, as these cases could have been miscoded terminations of pregnancy with signs of life at birth. However, after running the mortality analysis, it was noticed locally that there were survivors in EUROCAT registries at GA 23 weeks. For the morbidity studies, the exclusion criterion was lowered to be below 23 weeks.

2. Strength of linkage: the success of data linkage depended on the matching method and type of personal identifiers used. Where a national unique identifier (ID) was available (eg, Denmark, Finland) over 99% of cases were matched, but success rates were generally lower when intermediary databases and a combination of other identifiers (eg, names, postcodes) were required to establish a match, particularly if these were incomplete or incorrect. A standard way of evaluating confidence in a match had to be developed so that decisions on inclusion for analyses could be consistently made, in order to avoid bias.

3. GA groups: the GA at birth was categorised into <28 weeks, 28–31 weeks, 32–36 weeks and ≥37 weeks when analysing survival, but due to small numbers of survivors at under 28 weeks gestation, the two lowest GA categories were combined to <32 weeks.

4. Birth weight: birth weight was categorised into very low birth weight '<1500 g', low birth weight '≥1500 g to <2500 g', normal birth weight '≥2500 g to <4000 g' and high birth weight '≥4000 g'. Births <1000 g were not distinguished from those between 1000 and 1499 g as there were too few cases for the data to be analysed accurately.

5. Singletons versus multiples: there is uncertainty about whether the survival in twins with CAs is lower than that in singletons.[34–37] Hence, all survival analyses were performed on singletons alone and then multiples and singletons combined (multiples were not analysed alone as for many registries small numbers would limit the analyses that could be performed). This enables the survival of singletons and multiples in children with CAs to be analysed in detail. When examining morbidity, multiplicity was treated as one of the risk factors for increased risk of hospitalisations and lengths of stay to enable any association to be analysed, but with less detail than for survival. The majority of analyses included singletons and multiples combined.

6. Prenatal diagnoses: the GA at prenatal diagnoses was categorised into <22 weeks, 22–31 weeks, ≥32 weeks, GA not known and no prenatal diagnosis. For Finland,

**Table 3** EUROCAT congenital anomaly registries in EUROlinkCAT: start year, births in the population up to 2014, live births with an anomaly in the study period and ability to link to mortality, healthcare, prescription and education data

| Congenital anomaly registry | Planned start year | Actual start year | Number of live births with an anomaly | | Linkages occurring (Y), Not occurring (N) and Planned but no longer occurring (Y) | | | | | | |
|---|---|---|---|---|---|---|---|---|---|---|---|
| | | | Planned to be linked | Linked to mortality data by 8/2020 | Mortality | Hospital data for child | | Hospital data for mother | | Prescriptions | Education |
| | | | | | | In-patient | Out-patient | In-patient | Out-patient | | |
| Belgium: Antwerp | 1995 | 1997 | 8083 | 7865 | Y | N | N | N | N | N | N |
| Croatia: Zagreb | 1995 | 2011 | 2232 | 441† | Y | Y | Y | N | N | N | N |
| Denmark: Funen | 1995 | 1995 | 2418 | 2425 | Y | Y | Y | Y | Y | Y | Y |
| Finland | 1995 | 1995 | 44869 | 42861 | Y | Y | Y | Y | Y | Y | Y |
| France: Île de la Réunion | 2002 | NC | 3855 | NC | Y | N | N | N | N | N | N |
| France: Paris | 1997 | 1995 | 13335 | 11623 | Y | N | N | N | N | N | N |
| Germany: Saxony-Anhalt | 1995 | 2005 | 8821 | 8698 | Y | N | N | N | N | N | N |
| Italy: Emilia Romagna | 1995 | 2008 | 11447 | 7327 | Y | Y | N | N | N | Y | N |
| Italy: Tuscany | 1995 | 2005 | 9827 | 5187 | Y | Y | N | N | N | Y | Y |
| Malta | 1995 | 2005 | 2470 | 2718 | Y | N | N | N | N | N | N |
| Netherlands: Northern | 1995 | 2005 | 8567 | 8325 | Y | Y | Y | Y | Y | Y | N |
| Norway | 1999 | 1999 | 26938 | 27201 | Y | N | N | N | N | N | N |
| Portugal: South | 1995 | 2000* | 3425* | 2447* | N | Y | N | N | N | N | N |
| Spain: Basque Country | 1995 | 1995 | 4883 | 5904 | Y | Y | N | N | N | N | N |
| Spain: Valencian Region | 2007 | 2007 | 7438 | 7389 | Y | Y | N | N | N | Y | N |
| UK: East Midlands and South Yorkshire | 1998 | AL | 18549 | AL | Y | Y | N | N | N | N | Y |
| UK: Northern England | 2000 | AL | 8617 | AL | Y | Y | N | N | N | N | Y |
| UK: South West England | 2005 | AL | 11671 | AL | Y | Y | N | N | N | N | Y |
| UK: Thames Valley | 1995 | AL | 5142 | AL | Y | Y | N | N | N | N | Y |
| UK: Wales | 1998 | 1998 | 18239 | 18128 | Y | Y | Y | Y | Y | Y | Y |
| UK: Wessex | 1995 | AL | 7771 | AL | Y | Y | N | N | N | N | Y |
| Ukraine: West | 2005 | 2006 | 6166 | 5835 | Y | N | N | N | N | N | N |
| Total | | | 234763 | 155594† | 19 | 15 | 5 | 4 | 4 | 7 | 9 |

The registry in Basque was unable to complete the planned linkages to hospital data due to COVID-19.
*Values for WP4 morbidity linkage have been provided, as mortality linkage was never planned.
†The 441 cases in Zagreb were not included in analyses due to poor quality of the mortality linkage.
AL, awaiting linkage as of August 2020; NC, linkages could not be completed; Y, linkages are no longer planned as of August 2020.

**Table 4** Use of a reference population in morbidity and education analyses

| Congenital anomaly registry | Reference population |
|---|---|
| Croatia: Zagreb | Sample of children |
| Denmark: Funen | Whole population |
| Finland | Whole population |
| France: Île de la Réunion | Not provided |
| Italy: Emilia Romagna | Whole population |
| Italy: Tuscany | 10% of population |
| Netherlands: Northern | 10% of population |
| Portugal: South | Sample of children |
| Spain: Basque | Not provided |
| Spain: Valencian Region | Whole population |
| UK: East Midlands and South Yorkshire | Aggregate data from population for morbidity and population sample for education |
| UK: Northern England | Aggregate data from population for morbidity and population sample for education |
| UK: South West England | Aggregate data from population for morbidity and population sample for education |
| UK: Thames Valley | Aggregate data from population for morbidity and population sample for education |
| UK: Wales | Whole population |
| UK: Wessex | Aggregate data from population for morbidity and population sample for education |
| Ukraine: West | No longer in morbidity study |

the GA was often not recorded, only the trimester of diagnoses. Finland's first trimester diagnoses (week 0 to week 12) mapped exactly to the EUROlinkCAT <22 weeks category. It was decided that Finland's second trimester (week 13 to week 27) diagnoses were assumed to occur at 22–31 weeks and third trimester (week 28 onwards) diagnoses occurred at ≥32 weeks. These assumptions were also checked based on the distribution of those cases in Finland with a 'known' age at discovery and the assumptions held.

7. Length of stay (LOS): the LOS of the child in hospital was calculated after excluding the stay associated with the birth. Methods of identifying the birth stay varied in different countries. For hospital admissions where admission and discharge occurred on the same day, the LOS was considered to be 0.5 days. If an admission record was missing a discharge date, then discharge date=date of admission+2×(date of latest

**Table 5** Standardised variables from the EUROCAT database

| EDMP variables used (core variables are shaded in blue) | | |
|---|---|---|
| **Baby and mother** | | |
| 1 | CENTRE | Centre number |
| 2 | NUMLOC | Local ID of case |
| 3 | BIRTH_DATE | Date of birth |
| 4 | SEX | Sex |
| 5 | NBRBABY | Number of babies delivered |
| 6 | SP_TWIN | Specify twin type of birth, like or unlike, zygosity |
| 7 | NBRMALF | Number of malformed in multiple set |
| 8 | TYPE | Type of birth |
| 9 | CIVREG | Civil registration status |
| 10 | WEIGHT | Birth weight |
| 11 | GESTLENGTH | Length of gestation in completed weeks |
| 12 | SURVIVAL | Survival beyond 1 week of age |
| 13 | DEATH_DATE | Date of death |
| 14 | DATEMO | Date of birth of mother |
| 15 | AGEMO | Age of mother at delivery |
| 16 | BMI | Maternal body mass index |
| 17 | RESIDMO | Mother's residence code |
| **Diagnosis** | | |
| 19 | WHENDISC | When discovered |
| 20 | CONDISC | Condition at discovery |
| 21 | AGEDISC | If prenatally diagnosed, gestational age at discovery |
| 22 | FIRST PRE | First positive prenatal test |
| 24 | KARYO | Karyotype of infant/fetus |
| 25 | SP_KARYO | Specify karyotype |
| 26* | GENTEST | Genetic test |
| 27* | SP_GENTEST | Specify genetic test |
| 28 | PM | Postmortem examination |
| 29 | SURGERY | First surgery for malformation performed or planned |
| 30 | SYNDROME | Syndrome |
| 31 | SP_SYNDROME | Specify syndrome |
| 32 | MALFO1 | Malformation |
| 33 | SP_MALFO1 | Specify malformation |
| 34 | MALFO2 | As MALFO1 |
| 35 | SP_MALFO2 | Specify malformation |
| 36 | MALFO3 | As MALFO1 |
| 37 | SP_MALFO3 | Specify malformation |
| 38 | MALFO4 | As MALFO1 |
| 39 | SP_MALFO4 | Specify malformation |
| 40 | MALFO5 | As MALFO1 |
| 41 | SP_MALFO5 | Specify malformation |
| 42 | MALFO6 | As MALFO1 |
| 43 | SP_MALFO6 | Specify malformation |

Continued

## Table 5  Continued

| | EDMP variables used (core variables are shaded in blue) | |
|---|---|---|
| 44 | MALFO7 | As MALFO1 |
| 45 | SP_MALFO7 | Specify malformation |
| 46 | MALFO8 | As MALFO1 |
| 47 | SP_MALFO8 | Specify malformation |
| 57 | OMIM | OMIM code/type of Mendelian inheritance |
| **Exposure and family history** | | |
| 58 | ASSCONCEPT | Assisted conception (where available) |
| 59† | OCCUPMO | Mother's occupation at time of conception |
| **Sociodemographic** | | |
| 91 | MATEDU | Maternal education |
| 92* | SOCM | Socioeconomic status of mother |
| 93 * | SOCF | Socioeconomic status of father |
| 94 | MIGRANT | Migrant status |
| **Derived variables** | | |
| | Byear | Year of birth |
| | birth_type | Live birth, stillbirth, spontaneous abortion, TOPFA, not known definitions of stillbirths and spontaneous abortions vary between regions. This variable recodes birth type according to EUROCAT's specifications: cases with gestational age ≥20 weeks are recoded as 'stillbirths' (irrespective of the local definition of stillbirth/spontaneous abortion). |
| | casestatus | Only cases with casestatus=1 or 2 |
| | al1–al114 | EUROCAT subgroups: (0=No, 1=Yes). Based on EUROCAT coding in Guide 1.4 |
| | mult_malf | Algorithm for case classification into isolated and multiple anomalies |

*See work of the Standardisation Committee (viii).
†EUROCAT Guide 1.4 use ISCO-08 classifications.
EDMP, EUROCAT Data Management Programme; TOOPFA, termination of pregnancy for a fetal anomaly.

## Table 6  Coding of male or female of the live births in different linked databases in EUROlinkCAT

| Centre | Variable name | Code | |
|---|---|---|---|
| | | Male | Female |
| UK: Wales | DEC_SEX_CD | 1 | 2 |
| Germany: Saxony-Anhalt | EF306 | 1 | 2 |
| Finland | SUKUP | 1 | 2 |
| Italy: Tuscany | SESSO | 1 | 2 |
| France: Île de la Réunion | SexeDefunt | 1 | 2 |
| France: Paris | SexeDefunt | 1 | 2 |
| Netherlands: Northern | geslacht | 1 | 2 |
| Croatia: Zagreb | GENDER | M or 1 | F or 2 |
| Ukraine: West | CH_SEX | 1 | 2 |
| Belgium: Antwerp | SEX | 1 | 2 |
| Norway | KJONN | 1 | 2 |
| UK: England | sex | 1 | 2 |
| Spain: Basque Country | SEXO | 1 | 6 |
| Spain: Valencian Region | SEXO | 1 | 6 |
| Malta | gender | M | F |
| Italy: Emilia Romagna | SEX | M | F |
| Denmark: Funen | C_SEX | M | K |

procedure–date of admission). The date of discharge was set to the date of the child's 10th birthday or the end of the study period if it was after either of these two dates.

8. Socioeconomic status (SES): all registries had different variables that could be considered to be a measure of the mother's SES. The variables included maternal occupation, maternal education and index of multiple deprivation derived from residential codes at birth. Registries were asked to select the variable they believed was the most relevant and to recode their selected SES proxy variable into three groups of approximately equal proportion to enable comparing between, for example, mothers in the highest group to mothers in the lowest group. The effect of SES on survival would be analysed using Cox proportional hazard models within each registry. However, only seven registries were able to provide a proxy SES variable that was reasonably complete for some or all of the time period of the study. It was also planned to investigate the association between risk factors such as birth weight after adjusting for SES, but due to the lack of information on SES this was not included in further multivariable analyses.

9. Maternal country of birth: it was determined that the maternal country of birth variable would be used as a proxy for non-European ethnic origin, as we were aware that ethnic origin is poorly recorded. However, for those registries with reasonably complete data on this, almost 100% of children were reported as being of European ethnic origin. Therefore, this variable was not included in subsequent analyses as the number of children considered as 'non-European ethnic origin' was too small to analyse.

10. Cause of death: cause of death based on the death certificates was classified for deaths <1 year and for 1–9 years separately. Death related to preterm birth is very common in the first year after birth, but not as relevant to children at 1–9 years of age. Injuries and poisoning are more common after the first year. The main causes of deaths were classified into six groups for deaths <1 year and 11 groups for children aged 1–9 years. When working with the results tables it was clear that many of these classification groups included many small numbers and data could not be

| Patient data - 1 row per EUROCAT case | | | | | | | |
|---|---|---|---|---|---|---|---|
| Child ID | Child date of Birth | Child date of death | Type Matching | Variables relating to linkage | Variables relating to child | Variables relating to mortality | Variables relating to mother |

**Figure 1** Structure of mortality and EUROCAT data used for analysing children's survival.

extracted from the databases. For some registries it was only possible to give cause of death as either 'congenital anomaly' or else 'any other cause of death'.

11. Surgery: a number of different coding systems were used across registries to code surgeries and other procedures (eg, NCSP by NOMESCO, ICD-9-CM, OPCS-4). Frequency lists for all codes describing surgeries and other procedures were obtained from the linked datasets. Two paediatricians then independently determined if a code was a surgery or for another procedure and then a consensus between the two clinicians was reached over codes classified as codes for surgeries. Further subdivision into anomaly-specific surgeries was carried out for anomalies for which specific

surgeries could be identified that would be expected to be performed on these children.

12. Intensive care: it was planned to analyse the number of days in intensive care, however, only five registries could provide this. Therefore, only whether a child had ever been admitted to intensive care was analysed rather than their LOS.

13. Ventilation: it was planned to analyse the number of days on ventilation. However, as it was decided that the LOS in intensive care was not going to be analysed, the same decision was made for ventilation and only whether a child had ever been on ventilation was analysed.

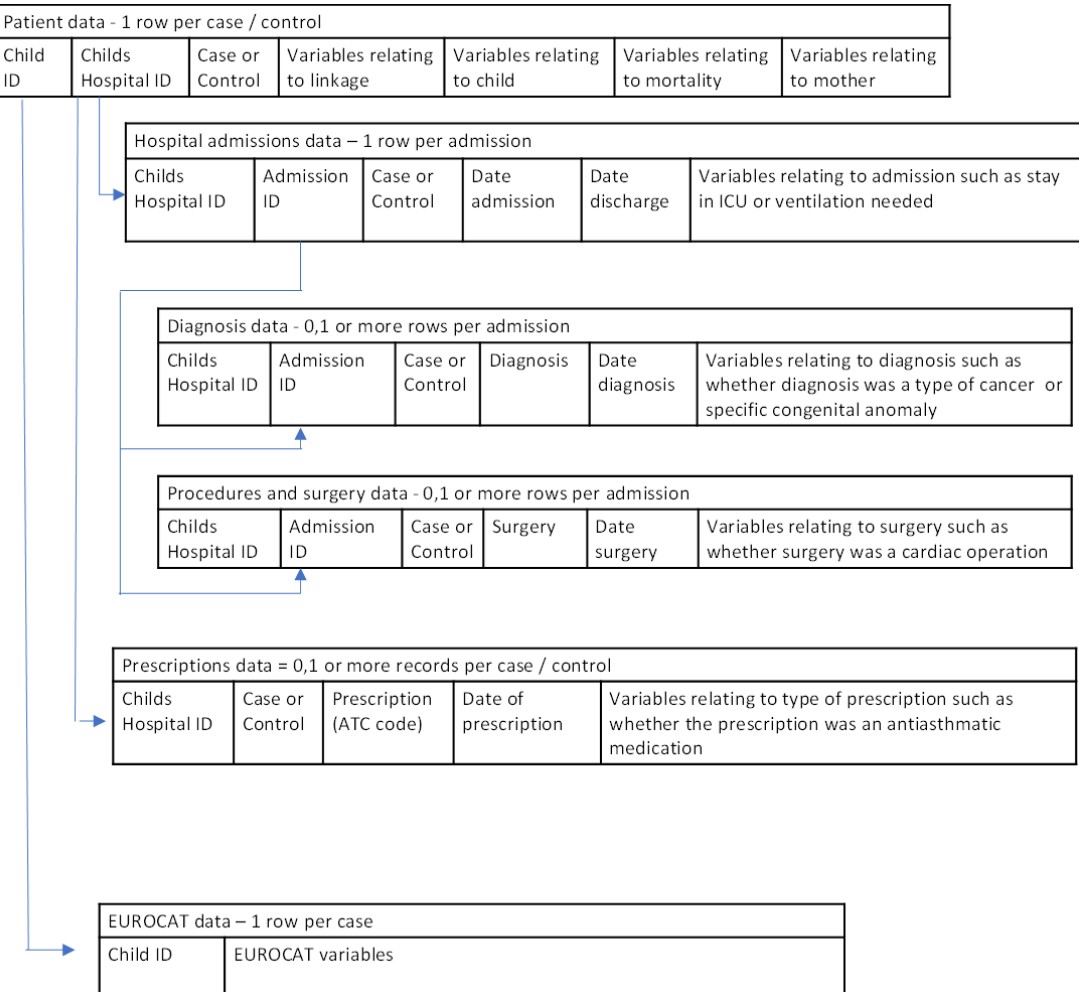

**Figure 2** Structure of hospital admissions, prescription data and EUROCAT data used for analysing children's morbidity. ATC code, Anatomical Therapeutic Chemical Classification System code; ICU, intensive care unit.

## Assessment of quality of linkage and quality of linked data

Many registries linked their data to National Vital Statistics, which are databases that record all live births with follow-up until the child dies or emigrates outside the country/region of interest. Therefore, for these registries for the survival analysis, any child whose record was not in the National Vital Statistics Database was judged to be a non-match and overall linkage could be assessed. Some registries were only able to link to death certificates which meant that a non-match, that is, no death certificate found, was assumed to indicate that the child was still alive. The data from these registries were only included in the survival analyses if there was additional information about the quality of the linkage. For example, in Malta, due to the small well-defined population, there was confidence that all deaths had been identified.

In some countries all national databases use the same unique ID number (eg, Finland). So, identifying a child in the National Vital Statistics meant that there was confidence that any hospital stays up to 10 years of age would also be identified. For other registries, as not all children were likely to be admitted to hospital, each case was searched for in, not only the in-patient hospital database (which included the mother's visit for the birth), but also any other healthcare databases (such as outpatient, primary-care or prescription databases) for longer than the 10 years of follow-up in the study. The lack of information in any healthcare database was judged to mean a non-match. Sensitivity analysis was performed to assess if there were differences in results if the non-matches were included.

For the education data, all children known to be alive were assumed to be included in the National Education Databases and therefore any case not identified was assumed to be a non-match.

Syntax scripts were developed centrally by St George's, University of London (SGUL) to evaluate the accuracy of the linkage and identify any factors leading to missed links (eg, deaths within the first week of life). For each registry, the proportion of births in any single year of data that have not been linked is calculated and the data from any year with less than 85% of cases linked will be excluded from further analyses. Second, the quality of linked data items was also evaluated: a variable that was >20% missing in a year would be excluded from any analysis in which it featured; and variables that were recorded by both the CA registry and the linked database would be compared for agreement, by year. In general, it was found that data quality was poorer in the earlier years and tended to improve over time; however, if data quality fluctuated across the years, then only the longest consecutive period where quality was above the threshold would be analysed.

## Statistical analysis

Protocols and syntax scripts are developed centrally to create aggregate data and perform specific analyses on the individual cases in each standardised data set in STATA (V.13 and upwards). This allows each register to submit aggregated data and analytical results (eg, Kaplan-Meier estimates, HRs and CIs), rather than individual case data, to the EUROlinkCAT central results repository (CRR) at UU, UK using a secure web platform. UU collates the aggregate data and results and provides these data to the researchers responsible for the different analyses and publications. Multi-centre European analyses will be performed by combining the individual registries' aggregated data and analytic results, using meta-analytic techniques. Additional work is required to develop suitable models for combining survival data from several registries when the sample sizes are very small as observed in many registries.

## Small number restrictions (statistical disclosure control)

Four countries have limitations on the release of aggregate data and analytic results if the numbers of births involved are very small (generally under eight births). This situation arises in many analyses involving specific CAs, as CAs are rare, with some affecting less than 1 in 10 000 live births. Solutions to enable the maximum amount of data to be included in all multi-centre European analyses varied according to country. The Northern Netherlands released data if all exported results were rounded to the nearest 5. Rounding all frequencies ensures that original numbers cannot be inferred. For Denmark, a few named researchers at SGUL and UU were allowed access to the aggregate data for the purpose of collating and including in pooled-analysis, on condition that it was securely stored and processed; that any individual results involving fewer than five people were not released; and that personal identification was not possible from any released results. The SAIL databank (Wales) provided data to the CRR with the requirement that aggregate data on fewer than five people were not released and could not be calculated from any information in the public domain. The registry from Antwerp, Belgium could not release any information on three or fewer cases.

## Patient and public involvement

A series of focus groups has been held in different European countries involving parents with a child with one of four predefined CAs with different health problems covering learning difficulties, physical disabilities, visible defects and non-visible defects with higher mortality. The four anomalies selected were: CHD requiring surgery (referred to as severe CHD—a usually non-visible defect with high mortality), cleft lip (a visible defect often with speech problems), spina bifida (a physical disability with associated incontinence problems) and DS (Trisomy 21; a visible defect with learning difficulties and often associated with CHD). The focus groups have investigated parental experiences of having a child with one of the above anomalies and assessed parental research priorities and a paper will be published in due course.

In addition, a European survey concerning the diagnosis, medical care, education and everyday life will be

distributed to parents across Europe with children with the same four CAs as described above. Registries will ensure the questions in the survey are appropriate for their country (eg, the provision of health services, given how this differs in various European countries) and will translate the survey into their native language with back translation to confirm the accuracy of the translation. The aim is for the survey to be distributed via social media by parent support groups across Europe to engage with a wide spectrum of parents.

## DISCUSSION

The aggregate data and results from the CRR in EUROlinkCAT will provide important information on the survival, morbidity and education of children born with a CA in Europe. Researchers in each CA registry will be encouraged to also perform specific local analysis, in order to fully exploit the research potential of linked datasets. The establishment of a method of standardising data from each registry linkage into a CDM provides valuable infrastructure enabling future multi-national studies to be performed in an efficient manner and new registries to become involved.

The strength of this study is that the researchers are a multidisciplinary group, many of whom have collaborated successfully for many years through being members of EUROCAT. In addition, EUROlinkCAT is able to build on all the standardisation procedures already established in EUROCAT. The implementation of a CDM enables the same centrally developed syntax script to be run in all the different registries which is efficient and also ensures standardisation of analysis across the registries. The use of a reference population when analysing healthcare data will aid in the identification of the source of differences between registries (eg, average LOS in hospitals will differ) and therefore enable us to better quantify the burden of disease attributable to CAs in each country. Comparisons of the accuracy of healthcare databases with respect to recording CA cases will be informative and enable improvements in those areas with less accurate data. We will also be developing recommendations on how to use the available healthcare data in an optimal way to provide information on children with CAs in areas without active CA registries.

One of the challenges of EUROlinkCAT is the ability of the CA registries to link their data to external data sources due to different local data information governance issues and the availability of suitable electronic healthcare databases. This requires flexibility in including registries in only specific subprojects and acceptance that not all registries may be able to perform the linkages planned. In addition, some registries require support from other partners in all aspects of the project, including applying for ethics permissions, adapting protocols, standardising data and running statistical syntax scripts. The restriction of not being able to share individual case data and also aggregate data that might be disclosive or identifiable

means that all analyses must be performed locally using a generic modelling strategy. This does limit the use of iterative procedures to explore data in detail. The major limitation to the study is that only specific areas in Europe are represented, with a lack of data in particular from Eastern Europe. Interpretation of differences across Europe is challenging as it will be essential to interpret results in the light of knowledge about the differences in healthcare and education practices across Europe.

The EUROlinkCAT project will enable important hypotheses concerning the survival, health and education of children with CAs in Europe to be investigated. The standardised methods and CDMs will all be available freely on the EUROlinkCAT website and will be available for use in future research projects to benefit from and build on this work, so as to enable other multi-centre European projects to exploit routine healthcare data available in Europe.

## ETHICS AND DISSEMINATION

The CA registries have the required ethics permissions and procedures for routine surveillance, data collection and transmission of anonymised data to the EUROCAT central database, according to national guidelines and they were required to submit evidence of these permissions to the EUROlinkCAT ethics portfolio. Local registries follow national legislation as to whether parental consent is needed for registration of babies with anomalies. Each registry was responsible for applying for and obtaining the additional ethics and other permissions (eg, data sharing agreements) required to link and analyse their data for EUROlinkCAT. This was an extremely lengthy process in some countries as the original data collection did not include expectation or consent for the data to be used in research, and a new legal basis had to be established. Additional assurances and procedures were adopted by registries (eg, publication of privacy notices) to ensure compliance with the General Data Protection Regulation which came into force on 25 April 2018 in the European Union. A checklist of minimum specifications for data storage/backup was completed by each registry. Three registries took over 3 years to get the ethics, legal basis, data protection, information governance and data sharing agreements in place. UU obtained ethics permission for the CRR.

Each registry participating in the focus groups with parents was responsible for ensuring the correct ethics approvals were in place. Similarly, the registries participating in the dissemination of the parents' survey will be responsible for ensuring the necessary ethics permissions are obtained.

An Ethics and Data Protection Advisory Board consisting of three independent advisors with the relevant expertise monitor all ethical considerations in this project.

The CRR will be used for multiple studies and the results from these will be disseminated in peer reviewed papers and conference presentations. It is hoped that the

experience gained with distributing the parents' survey using parent support groups and social media will also lead to development of a framework to enable dissemination of results to be made more directly to parents. In addition, a series of reports will be written including recommendations for improving the collection and analysis of data on CAs in routinely collected data in the healthcare databases.

**Author affiliations**
[1]Population Health Research Institute, St George's University of London, London, UK
[2]Paediatric Department, Hospital Lillebaelt, Kolding, Denmark
[3]Faculty of Life and Health Sciences, Ulster University, Coleraine, UK
[4]Children's Hospital Zagreb, Centre of Excellence for Reproductive and Regenerative Medicine, Medical School University of Zagreb, Zagreb, Croatia
[5]Biomedical Computing Limited, Battle, UK
[6]Polish Registry of Congenital Malformations, Chair and Department of Medical Genetics, Poznan University of Medical Sciences, Poznan, Poland
[7]IMER Registry (Emila Romagna Registry of Birth Defects), University Hospital of Ferrara, Emilia-Romagna, Italy
[8]Instituto di Fisiologia Clinica, Consiglio Nazionale delle Ricerche, Pisa, Italy
[9]Population Health Sciences Institute, Newcastle University, Newcastle upon Tyne, UK
[10]Malformation Monitoring Centre Saxony-Anhalt, Otto von Guericke University Medical Faculty, Magdeburg, Germany
[11]Department of Genetics, University Medical Centre Groningen, Groningen, The Netherlands

**Collaborators**  All below named members of the EUROlinkCAT Consortium have contributed to the development of the protocol with their knowledge and experience of the surveillance of congenital anomalies. Names are given in Consortium organisation numerical order: Population Health Research Institute, St George's, University of London, UK: Abigail Reid, Liz Limb, Gillian Briggs, Nicholas Connor; Hospital Lillebaelt, Region Syddanmark, Denmark and Section of Epidemiology, University of Copenhagen, Denmark: Stine Kjaer Urhoj; Hospital Lillebaelt, Region Syddanmark, Denmark: Mads Damkjær, Christina Neergaard Pedersen; Population Health Sciences Institute, Faculty of Medical Sciences, Newcastle University, Newcastle upon Tyne, UK: Svetlana V. Glinianaia; Emila Romagna Registry of Birth Defects (IMER), University of Ferrara, Italy: Gianni Astolfi, Annarita Armaroli; Neonatal Intensive Care Unit, IMER Registry, Department of Medical Sciences, University Hospital of Ferrara, Italy: Elisa Ballardini; Territorial Care Service - Emilia Romagna Health Authority, Italy: Aurora Puccini; Childrens Hospital Zagreb, Croatia: Ljubica Boban; Institute of Clinical Physiology, National Research Council (Consiglio Nazionale delle Ricerche-Istituto di Fisiologia Clinica), Pisa, Italy: Silvia Baldacci, Alessio Coi, Lorena Mezzasalma, Michele Santoro; University of Groningen, University Medical Center Groningen, Department of Genetics, Groningen, the Netherlands: L. Renée Lutke, Nicole H.A. Siemensma-Mühlenberg; Public Health Wales National Health Service Trust, Wales, UK: David Tucker; Institut National de la Santé et de la Recherche Médicale, Paris, France: Babak Khoshnood, Nathalie Bertille, Nathalie Lelong, Makan Rahshenas; Rare Diseases Research Unit, Foundation for the Promotion of Health and Biomedical Research in the Valencian Region, Valencia, Spain: Clara Cavero Carbonell, Laia Barrachina Bonet, Laura Garcia Villodre; Rare Diseases Research Unit, Foundation for the Promotion of Health and Biomedical Research in the Valencian Region, Valencia, Spain and Public Health and Preventive Medicine Department, University of València, Spain and Public Health Regional Health Administration (DG Salud Publica y Adicciones), Generalitat Valenciana, Valencia, Spain: Óscar Zurriaga; Polish Registry of Congenital Malformations, Department of Medical Genetics, Poznan University of Medical Sciences. Poznan, Poland: Anna Jamry-Dziurla; Finnish Institute for Health and Welfare, Finland: Sonja Kiuru-Kuhlefelt, Mika Gissler, Anna Heino; OMNI-Net for Children International Charitable Fund, Rivne, Ukraine: Wladimir Wertelecki; OMNI-Net for Children International Charitable Fund, Rivne, Ukraine and Khmelnytsky City Children's Hospital, Khmelnytsky, Ukraine: Diana Akhmedzhanova, Nataliia Zymak-Zakutnia; OMNI-Net for Children International Charitable Fund, Rivne, Ukraine and Rivne Regional Medical Diagnostic Center, Rivne, Ukraine: Lyubov Yevtushok, Lyubov Ostapchuk, Oksana Tsizh; OMNI-Net for Children International Charitable Fund, Rivne, Ukraine and Volyn Regional Children's Territorial Medical Center, Lutsk, Ukraine: Serhii Lapchenko; Epidemiology Department, National Registry of Congenital Anomalies, National Institute of Health Doctor Ricardo Jorge (Instituto Nacional de Saúde Doutor Ricardo Jorge), Lisbon, Portugal: Carlos Matias Dias, Ausenda Machado, Ana João Santos, Liliana Antunes, Paula Braz; Register of Congenital Malformations, Centre Hospitalier Universitaire de La Réunion, Île de la Réunion, France: Hanitra Randrianaivo-Ranjatoelina, Benedicte Bertaut Nativel; Provinciaal Instituut voor Hygiëne (PIH), Antwerp, Belgium: Vera Nelen, Guy Thys, Elly Den Hond; Departamento de Salud Gobierno Vasco, Basque Country, Spain: Olatz Mokoroa Carollo; Swansea University, Wales, UK: Daniel Thayer, Ieuan Scanlon, Sue Jordan; Department of Global Public Health and Primary Care, University of Bergen, Bergen, Norway and Division of Mental and Physical Health Norwegian Institute of Public Health, Bergen, Norway: Kari Klungsøyr; Directorate for Health Information and Research, G'Mangia, Malta: Miriam Gatt.

**Contributors**  JKM is the principal author and the project's scientific coordinator. ML is the data coordinator. EG is the clinical coordinator. JD, AN, AP, JR and HdW are Steering Committee and Standardisation Committee members with JKM, ML and EG. IB and AL-B are Steering Committee members. AR is a Standardisation Committee member. JT is the senior statistician. JEG the research fellow. HC is the project manager. All authors contributed to, read and approved the final manuscript.

**Funding**  This project has received funding from the European Union's Horizon 2020 Research and Innovation programme under grant agreement No 733001 for 5 years: 1 January 2017 to 31 December 2021. The views presented here are those of the authors only, and the European Commission is not responsible for any use that may be made of the information presented here.

**Competing interests**  None declared.

**Patient consent for publication**  Not required.

**Provenance and peer review**  Not commissioned; externally peer reviewed.

**ORCID iDs**
Joan K Morris http://orcid.org/0000-0002-7164-612X
Ester Garne http://orcid.org/0000-0003-0430-2594
Maria Loane http://orcid.org/0000-0002-1206-3637
Ingeborg Barisic http://orcid.org/0000-0002-9085-6747
Anna Latos-Bieleńska http://orcid.org/0000-0002-9152-1674
Anna Pierini http://orcid.org/0000-0003-3321-9343
Judith Rankin http://orcid.org/0000-0001-5355-454X
Anke Rissmann http://orcid.org/0000-0002-9437-2790
Hermien de Walle http://orcid.org/0000-0002-1418-8281
Joachim Tan http://orcid.org/0000-0003-0462-4761
Joanne Emma Given http://orcid.org/0000-0003-4921-1944
Hugh Claridge http://orcid.org/0000-0001-5998-2860

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
