## [Reviewer comments · BMJ Open]

ARTICLE DETAILS

TITLE (PROVISIONAL)	EUROlinkCAT Protocol for a European population-based data linkage study investigating the survival, morbidity and education of children with congenital anomalies
AUTHORS	Morris, Joan; Garne, Ester; Loane, Maria; Barisic, Ingeborg; Densem, James; Latos-Bieleńska, Anna; Neville, Amanda; Pierini, Anna; Rankin, Judith; Rissmann, Anke; de Walle, Hermien; Tan, Joachim; Given, Joanne; Claridge, Hugh

VERSION 1 – REVIEW

REVIEWER	Ayşe Şener Taplak Yozgat Bozok Universitesi
REVIEW RETURNED	13-Feb-2021

GENERAL COMMENTS	This is an important study aimed to provides information on the survival, morbidity, and education of children born with a CA in Europe. In addition, the study was aimed to provide a data source for developing recommendations on how to use the available health care data in an optimal way to provide information on children with CAs in areas without active CA registries. In the study, the reference group was chosen to identifying the source of differences between registries. The protocol creation process, justifications, and included criteria were explained in the manuscript. This is a well-written article with all these dimensions. References older than ten years should be updated if possible.
---

REVIEWER	Linda Wijlaars UCL, Population, Policy and Practice
REVIEW RETURNED	25-Feb-2021

GENERAL COMMENTS	The authors present a protocol for a very exciting European linkage study, linking congenital anomaly registry data to administrative routinely collected mortality and hospital data (and education data, though unfortunately this wasn't able to fit a common data model across participating countries). The protocol is well written and clear. I just have a few minor corrections and suggestions for clarifications. Minor comments: 1) There is a typo in the first sentence: "Over 130,000 children born in Europe every year will have major a CA". 'Major' and 'a' should be the other way around. 2) In Table 1, there are some symbols indicating footnotes in the
---

	header and first rows (for all anomalies, ICD-9-BPS and codes D1810 and 2281), however, no footnotes seem to be present (unless I am just looking in the wrong place). Could the authors add these? 3) Table 3: perhaps this is a non-issue that will be solved once BMJ Open applies its house style to the table, but it would be helpful to have banded rows (or borders maybe every 5 rows or so?). As the table is quite wide, it is difficult to determine which value in the columns on the left corresponds to which country. 4) Table 5 appears to be missing footnotes as well. Perhaps a footnote could be added to the socioeconomic status variables to refer to note viii under the work of the standardisation committee section? 5) In the 'Assessment of Quality of Linkage and Quality of Linked Data' the authors state data from any year with less than 85% of cases linked will be excluded from further analyses. Which data does this pertain to? The mortality data will not reach 85% linkage, and even for hospital admissions (especially in countries where hospital births are less common) not all children with a major CA will require hospital admission. For instance, Zylbersztejn et al (Phenotyping congenital anomalies in administrative hospital records, Paediatric and perinatal epidemiology 2020), report <80% of children with a CA had at least one hospital admission in the two years after birth. . A child not linking to a hospital record or mortality record could be correct rather than a missed link. This will be complicated as the linkage will differ by country. 6) For the 'Ethics and dissemination' section: the authors state this was a harrowing process. If possible, could they include some information on time lines? This is just a suggestion, but as a researcher working with linked administrative data (or trying to, more often) I find researchers unfamiliar with this field are often unaware of just how long it can take to get the, ethics, legal basis, data protection, information governance and data sharing agreements in place. Or it might make for a nice short research letter, comparing experiences in different European countries?
--	---

VERSION 1 – AUTHOR RESPONSE

Reviewer: 1

Dr. Ayşe Şener Taplak, Yozgat Bozok Universitesi Comments to the Author:

This is an important study aimed to provide information on the survival, morbidity, and education of children born with a CA in Europe. In addition, the study was aimed to provide a data source for developing recommendations on how to use the available health care data in an optimal way to provide information on children with CAs in areas without active CA registries. In the study, the reference group was chosen to identify the source of differences between registries. The protocol creation process, justifications, and included criteria were explained in the manuscript. This is a well-written article with all these dimensions. References older than ten years should be updated if possible.

Best Regards

There were 9 references that were older than ten years. We have updated 5 of them and deleted 3 of them which were unnecessary. We wish to retain one older reference as it is an important paper in the field.

Reviewer: 2

Dr. Linda Wijlaars, UCL, University College London Comments to the Author:

The authors present a protocol for a very exciting European linkage study, linking congenital anomaly registry data to administrative routinely collected mortality and hospital data (and education data, though unfortunately this wasn't able to fit a common data model across participating countries). The protocol is well written and clear. I just have a few minor corrections and suggestions for clarifications.

Minor comments:

1) There is a typo in the first sentence: "Over 130,000 children born in Europe every year will have major a CA". 'Major' and 'a' should be the other way around.

This has been corrected

2) In Table 1, there are some symbols indicating footnotes in the header and first rows (for all anomalies, ICD-9-BPS and codes D1810 and 2281), however, no footnotes seem to be present (unless I am just looking in the wrong place). Could the authors add these?

Apologies the footnotes have now been added

3) Table 3: perhaps this is a non-issue that will be solved once BMJ Open applies its house style to the table, but it would be helpful to have banded rows (or borders maybe every 5 rows or so?). As the table is quite wide, it is difficult to determine which value in the columns on the left corresponds to which country.

We have added shaded bands

4) Table 5 appears to be missing footnotes as well. Perhaps a footnote could be added to the socioeconomic status variables to refer to note viii under the work of the standardisation committee section?

Apologies the footnotes have now been added and an additional one for the socioeconomic status variable as suggested

5) In the 'Assessment of Quality of Linkage and Quality of Linked Data' the authors state data from any year with less than 85% of cases linked will be excluded from further analyses. Which data does this pertain to? The mortality data will not reach 85% linkage, and even for hospital admissions (especially in countries where hospital births are less common) not all children with a major CA will require hospital admission. For instance, Zylbersztejn et al (Phenotyping congenital anomalies in administrative hospital records, Paediatric and perinatal epidemiology 2020), report <80% of children with a CA had at least one hospital admission in the two years after birth. A child not linking to a hospital record or mortality record could be correct rather than a missed link. This will be complicated as the linkage will differ by country.

The text concerning in this section was not clear and we have added additional paragraphs (see below and inserted in text) on page 18 into describe when linkage was judged to have occurred.

Many registries linked their data to National Vital Statistics, which are databases that record all live births with follow up until the child dies or emigrates outside the country/region of interest. Therefore, for these registries for the survival analysis, any child whose record was not in the National Vital Statistics Database was judged to be a non-match and overall linkage could be assessed. Some registries were only able to link to death certificates which meant that a non-match, i.e. no death certificate found, was assumed to indicate that the child was still alive. The data from these registries were only included in the survival analyses if there was additional information about the quality of the linkage. For example, in Malta, due to the small well-defined population, there was confidence that all deaths had been identified.

In some countries all national databases use the same unique ID number (for example Finland). So, identifying a child in the National Vital Statistics meant that there was confidence that any hospital stays up to ten years of age would also be identified. For other registries, as not all children were likely to be admitted to hospital, each case was searched for in, not only the in-patient hospital database (which included the mothers visit for the birth), but also any other health care databases

(such as out-patient, primary-care or prescription databases) for longer than the ten years of follow-up in the study. The lack of information in any health care database was judged to mean a non-match. Sensitivity analysis was performed to assess if there were differences in results if the non-matches were included.

For the education data, all children known to be alive were assumed to be included in the National Education Databases and therefore any case not identified was assumed to be a non-match.

6) For the 'Ethics and dissemination' section: the authors state this was a harrowing process. If possible, could they include some information on time lines? This is just a suggestion, but as a researcher working with linked administrative data (or trying to, more often) I find researchers unfamiliar with this field are often unaware of just how long it can take to get the, ethics, legal basis, data protection, information governance and data sharing agreements in place. Or it might make for a nice short research letter, comparing experiences in different European countries?

We agree with the reviewer and have added text to indicate that three registries took over three years to obtain all the necessary agreements (page 21)